# Quality Assessment Assistance of Lateral Knee X-rays: A Hybrid Convolutional Neural Network Approach

**Simon Lysdahlgaard** [1,2,3], **Sandi Baressi Šegota** [4,*], **Søren Hess** [1,2,3], **Ronald Antulov** [1,2,3], **Martin Weber Kusk** [1,2,3] **and Zlatan Car** [4]

1 Department of Radiology and Nuclear Medicine, University Hospital of South West Jutland, University Hospital of Southern Denmark, Finsensgade 35, 6700 Esbjerg, Denmark; simon.lysdahlgaard@rsyd.dk (S.L.); soren.hess@rsyd.dk (S.H.); ronald.antulov@rsyd.dk (R.A.); martin.weber.kusk@rsyd.dk (M.W.K.)

2 Department of Regional Health Research, Faculty of Health Sciences, University of Southern Denmark, Campusvej 55, 5320 Odense, Denmark

3 Imaging Research Initiative Southwest (IRIS), Hospital of South West Jutland, University Hospital of Southern Denmark, 6700 Esbjerg, Denmark

4 Faculty of Engineering, University of Rijeka, Vukovarska 58, 51000 Rijeka, Croatia; car@riteh.hr

\* Correspondence: sbaressisegota@riteh.hr; Tel.: +385-51-505-715

**Abstract:** A common issue with X-ray examinations (XE) is the erroneous quality classification of the XE, implying that the process needs to be repeated, thus delaying the diagnostic assessment of the XE and increasing the amount of radiation the patient receives. The authors propose a system for automatic quality classification of XE based on convolutional neural networks (CNN) that would simplify this process and significantly decrease erroneous quality classification. The data used for CNN training consist of 4000 knee images obtained via radiography procedure (KXE) in total, with 2000 KXE labeled as acceptable and 2000 as unacceptable. Additionally, half of the KXE belonging to each label are right knees and left knees. Due to the sensitivity to image orientation of some CNNs, three approaches are discussed: (1) Left-right-knee (LRK) classifies XE based just on their label, without taking into consideration their orientation; (2) Orientation discriminator (OD) for the left knee (LK) and right knee (RK) analyses images based on their orientation and inserts them into two separate models regarding orientation; (3) Orientation discriminator combined with knee XRs flipped to the left or right (OD-LFK)/OD-RFK trains the models with all images being horizontally flipped to the same orientation and uses the aforementioned OD to determine whether the image needs to be flipped or not. All the approaches are tested with five CNNs (AlexNet, ResNet50, ResNet101, ResNet152, and Xception), with grid search and k-fold cross-validation. The best results are achieved using the OD-RFK hybrid approach with the Xception network architecture as the classifier and ResNet152 as the OD, with an average AUC of 0.97 ($\pm$0.01).

**Keywords:** artificial neural networks; image classification; machine learning algorithm; radiography; hybrid intelligent systems

**MSC:** 68T01

## 1. Introduction

One of the key issues in modern radiography is determining the quality of the obtained X-ray examinations (XE). XEs are acquired by radiographers, and any quality issue may not be noticed until the patient's XE reaches image interpretation. If the XE is not of satisfactory quality, it will be rejected and the patient re-examined. Re-examination can cause additional stress to the patient, increase the hospital workload, and may cause delays in patients' treatment. For these reasons, the number of low-quality XEs submitted for reading must be kept at a minimum. One way to accomplish this is to create an automatic system that will

classify the images into two groups—acceptable or unacceptable quality. Knee XE (KXE) are among those with the highest rejection rates [1], making them a relevant candidate for quality improvement by such a method.

Machine learning (ML) algorithms are a subset of artificial intelligence methods [2]. When it comes to image-based datasets, one of the most applied ML algorithms is the so-called convolutional neural network (CNN). CNNs utilize the convolution between the input image and internal filters, where values of filters are then adjusted through the training process, to achieve high-quality models. Mahum et al. (2021) [3] demonstrated the application of deep CNNs for feature extraction of XE focused on knee osteoarthritis detection. The authors applied CNN feature extraction with local binary patterns for texture features and histograms of oriented gradientsfor low-level features. The authors applied multiple ML algorithms to classify XE into four classes according to the Kellgren–Lawrence system and achieved 97% accuracy in detecting osteoarthritis. Another automated system for knee osteoarthritis detection was described by Feng et al. (2021), where a hybrid approach using a channel attention module and a spatial attention module was applied, achieving an F1 score of 0.6755 and an accuracy of 0.7023. Urban et al. (2020) [4] applied deep learning for shoulder implant classification on XE. The goal was to classify the prosthesis state after total shoulder arthroplasty. The authors compared the performance of deep learning CNNs with other classifiers when the out-of-domain data are used to pre-train models. CNN classifiers achieved an accuracy of approximately 0.8 in comparison to 0.56 or less achieved by other classifiers. Lately, CNNs were applied to XE for the detection and classification of COVID-19-related lung changes. Islam et al. (2020) [5] applied custom CNN learning algorithms in the detection of COVID-19 in XE, with similar work performed by Hira et al. (2021) [6], Momeny et al (2021) [7], Lorencin et al. (2021) [8], and Heidari et al. (2020) [9]. Sharma et al. (2021) demonstrated the use of CNNs to perform feature extraction and classification of XE in pneumonia differentiation. The varied applicability of CNNs has shown great accuracy in clinical tasks related to XE image classification, achieving accuracy scores between 0.67 and 0.99 [4–8]. The need to reduce the number of XEs performed on patients is noted by several researchers. Porter et al. (2021) [10] discussed the application of a quality improvement initiative to lower the number of inpatient post-thoracic surgery chest XE. The authors noted many issues of X-ray over-use due to lower-quality XE. However, the radiation exposure to individuals and groups should be kept "As low as reasonably achievable", consistent with the provision of the benefits of radiation used in society [11]. While there aremany articles discuss CNN application with XE, and many health professionals note the need for quality control in the field of XE [12] there are currently few articles that attempt to apply CNNs to this specific problem.

*Related Work*

Zhang et al. (2021) [13] apply the Feature Extraction CNN to the problem of fetal sonograph quality assessment. On the balanced dataset of 14,700 images, the authors manage to achieve AUC and F1 scores of 0.96. Lauermann et al. (2019) [14] describe the use of a customized deep learning algorithm, based on CNNs, for the problem of optical coherence tomography angiography quality assessment. The models are evaluated using accuracy and achieve a score of 0.97 on the validation dataset. Jalaboi et al. (2023) [15] focus on the application of a custom CNN architecture "ImageQX". The architecture is trained on 36,509 images, out of which the validation set consists of 9874 photographs. With the architecture at hand, authors achieve the score of $0.73 \pm 0.01$. Quality assessment on eye fundus photography is performed by Karlsson et al. (2021) [16]. They apply a random forest model on features extracted with Fourier transformation on a publicly available DRIMDB dataset consisting of 216 images. The authors demonstrate an accuracy score of 0.981, sensitivity score of 0.993, and specificity score of 0.958. Mairhofer et al. (2021) [17] extract the region of interest and classify the quality into one of three classes using a customized CNN-based framework. When evaluating the set of 950 images, the authors achieve an accuracy score of 0.94 with their proposed three-step framework. Chabert et al.

(2021) [18] apply multiple methods (linear discriminant analysis, quadratic linear analysis, support vector machine, logistic regression, and multilayer perceptron) for assessment of Lumbar MRI quality. The best results are achieved with a combination of methods, with a recall of 0.82 and an AUC of 0.97. Coyner et al. (2019) [19] demonstrate the use of custom deep CNNs for quality assessment of retinopathy images. Using a dataset of 4000 images, the authors achieve an AUC of 0.97, with a sensitivity of 0.94 and specificity of 0.84. Czajowska et al. (2022) [20] apply a hybrid system consisting of a deep CNN and a fuzzy reasoning system on quality assessment of author-collected dataset of high-frequency facial ultrasounds. The achieved results show that the proposed system achieves a classification accuracy of 0.92. The summary of the results for these related studies is given in Table 1.

**Table 1.** The main results of the related works, along with the number of images used in the study, as well as the main research topic/type of the images for which the quality assessment is performed (P—Precision, R—Recall, ACC—Accuracy, Spec.—Specificity, Sens.—Sensitivity).

| Ref. | Quality Assessment Type | Images | Scores |
|---|---|---|---|
| [13] | Fetal sonograph | 14,700 | P = 0.97, Sens. = 0.95, ACC = 0.94, F1 = 0.96, Spec. = 0.94, AUC = 0.96 |
| [14] | Angiography imagery | 200 | ACC = 0.97 |
| [15] | Teledermatological photography | 39,509 | F1 = $0.73 \pm 0.01$ |
| [16] | Fundus photography | 216 | ACC = 0.98, Sens. = 0.99, Spec. = 0.95 |
| [17] | Ankle radiography | 950 | ACC = 0.94 |
| [18] | Lumbar MRI | 95 | R = 0.82, AUC = 0.77 |
| [19] | Retinopathy images | 4000 | AUC = 0.97, Sens. = 0.94, Spec. = 0.84 |
| [20] | Facial skin ultrasounds | 17,425 | ACC = 0.92 |

There are a few knowledge gaps that can be identified by observing the related work. First, there is a clear lack of application on the specific issue of KXE quality assessment. Second, only a few studies consider the application of existing, tried, and tested CNN architectures and opt for a customized CNN approach. This approach complicates the studies through the introduction of unknown variables, as well as making them harder to reproduce, while they may not provide a benefit. Most of the research does not cross-validate the data and uses the standard train–test approach. Most authors use images as-is, without attempting to identify if there are performance benefits from using different-sized images. It is also important to note the number of images used, which range wildly—from fewer than a hundred images to tens of thousands, with the dataset used in the presented study falling around the middle at 4000 images. Finally, little testing is performed with combinations of various ML-based architectures—despite research in other areas showing that these types of methods can yield significantly improved results, only a single paper identified in the related work review applies them for quality assessment [20].

The main motivation of the presented work is to simplify the process of KXE examinations for both the patients and the medical staff. The low-quality KXEs refer to those KXEs in which the patient's knee is not positioned properly or is misoriented, not allowing for the medical examiner to diagnose the injury properly. When this happens, a new XE must be performed, which puts undue stress on the hospital system as well as lowering the quality of healthcare obtained by the patient, due to the delay introduced before a diagnosis can be made. This paper aims to apply CNNs for quality control of XE, specifically of lateral KXE, which are presented in two lateralities (left and right). Furthermore, we will determine whether classifying the images of different lateralities individually will provide better results compared to the dataset where both lateralities were mixed. The contributions of the presented work can be presented as:

- Developing an AI-based system for the classification of lateral KXE quality, in order to inform the technician performing the examination immediately if the positioning of the KXE is not satisfactory;
- Testing approaches combining different CNN architectures in various hybrid setups and comparing the performance to a standard, single-CNN approach in order to determine the need for the use of advanced CNN approaches; and
- Determining which parameters of the entire system generate the best performance, in regards to CNN architecture, a hybrid combination of the CNNs, hyperparameters, and image size, in order to enable simpler future development of models focused on similar issues.

## 2. Materials and Methods

The approach used to collect and prepare the dataset, as well as the training and evaluation process of models created using CNN algorithms will be described in this section. The entire presented methodology can be observed in Figure 1.

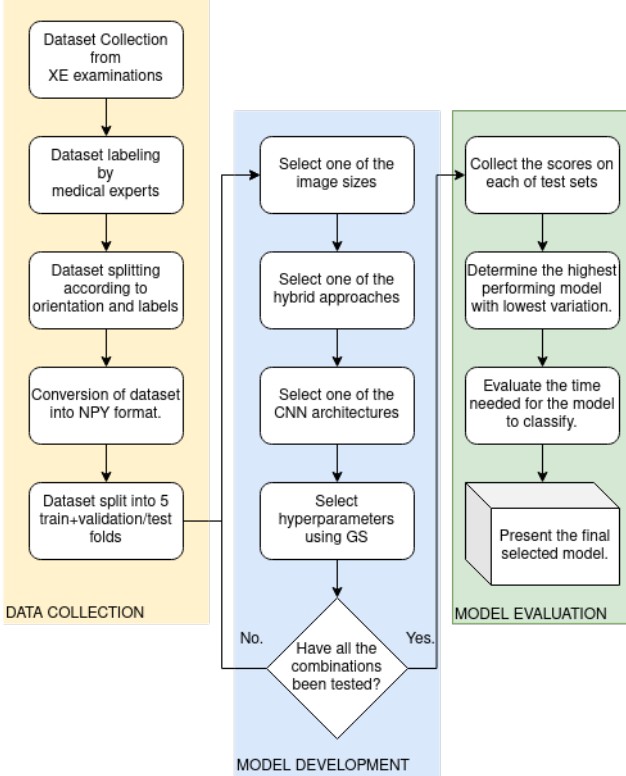

**Figure 1.** An overview of the methodology.

As Figure 1 shows, the methodology consists of three main parts—the dataset collection, model development, and model evaluation. The dataset collection was performed by medical experts and consisted of performing 2000 KXEs, labeling and splitting them, before converting them to an appropriate format for machine learning methodology to be applied. This was the second step, which mainly consisted of repeating the training process using the grid search (GS) hyperparameter variation approach and five-fold cross-validation on various image sizes and CNN approaches, which will be described going further. Finally, each of the models was evaluated, with the best-performing model being selected for further evaluation based on time performance.

### 2.1. Dataset Description

The dataset was collected at the University Hospital of South West Jutland in Esbjerg, Denmark, through the standard diagnostic procedure, using a Ysio Max Digital Radiogra-

phy system (Siemens Healthineers, Erlangen, Germany). Patients were positioned standing (weight-bearing), with the lateral side of the knee towards the image detector. The dataset consists of 4000 lateral knee XE, of which half are right knee (RK) XE and half are left knee (LK) XE. The data labeling of the XE as "acceptable" and "unacceptable" was performed by two reporting radiographers and used as the reference standard. Within each of the RK and LK datasets, half of the XE were labeled as "acceptable" and half as "unacceptable", for 2000 XE of each class in total. An example of the XE in the dataset is given in Figure 2.

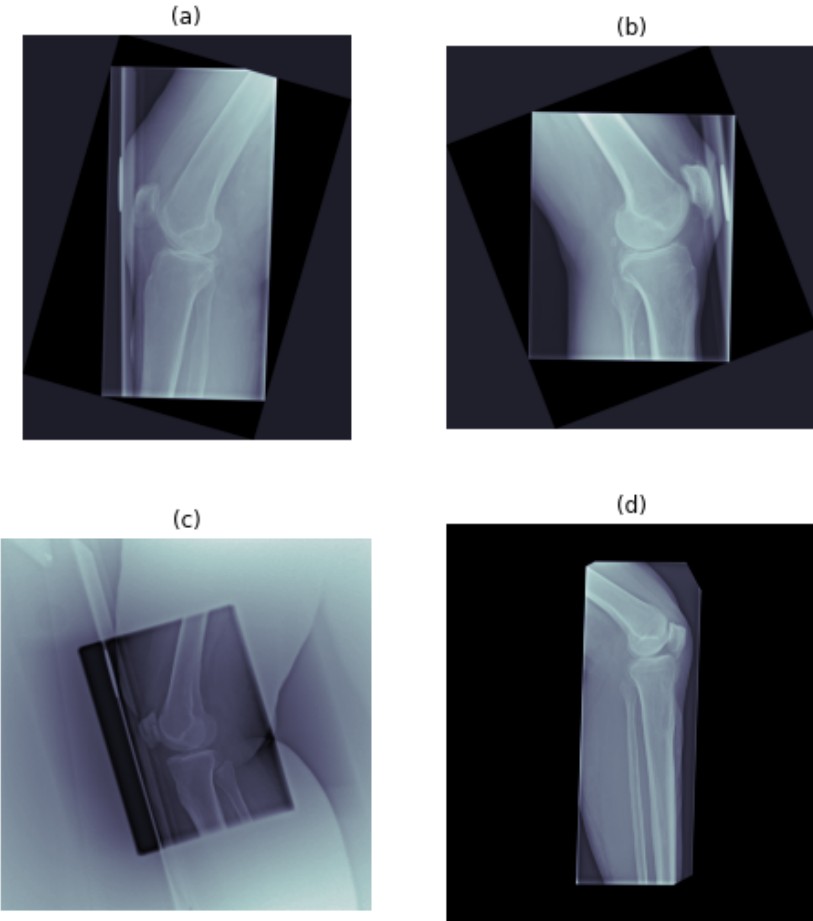

**Figure 2.** Example of used XE, per each class: (**a**) right-oriented, acceptable; (**b**) left-oriented, acceptable; (**c**) right-oriented, unacceptable; and (**d**) left-oriented, unacceptable.

*2.2. Dataset Preparation*

The XE were provided in the Digital Imaging and Communications in Medicine (DICOM) format. All XE were transformed from DICOM-format to NPY, which is readable for CNN architectures [21]. After the transformation, it was important to ensure that all the images were appropriately resized. The standard size used for image classification tasks has a width and height of $224 \times 224 \times 1$ [22,23]. As all the collected images have a width and height above 3000 and 2000 pixels, respectively, the original images were also resized to image sizes of $448 \times 448 \times 1$ and $896 \times 896 \times 1$, to test whether there will be a difference regarding classification performance. Single-channel images were used because the provided DICOMs contain single-channel, grayscale pixel data. Five separate datasets were prepared:

- Left and right knee XE (LRK);
- Left knee XE (LK);
- Right knee XE (RK);
- Left knee XE, with right knee flipped around the vertical axis (LFK); and

- Right knee XE, with left knee XE flipped around the vertical axis (RFK).

Each of these datasets is shaped as:

$$[N \times w \times h \times c \times L],\tag{1}$$

where:

- $N$ is the number of images in each dataset (2000 for LK and RK datasets, 4000 for LRK, LFK, and RFK datasets);
- $w$ is the image width (equal to either 223, 448, or 896);
- $h$ is the image height, always equal to $w$;
- $c$ is the number of image channels, always equal to 1; and
- $L$ is the binary image label, either "0" (unacceptable) or "1" (acceptable).

For training purposes of additional classification models, an additional dataset was created, which will be referred to as the orientation discrimination dataset (OD). It consists of all original XE, but the labels acceptable/unacceptable are not given. Instead, the labels are equal to knee orientations (as either left knee or right knee).

### 2.3. Machine Learning Methodology

Classification was performed using three different CNN types. A brief description of the used algorithms will be given in this section, with a prior description of the methodological approach. Hyperparameter tuning was performed for three different hyperparameters, including the solver, the number of epochs, and the batch size. The possible values of hyperparameters are given in Table 2.

**Table 2.** Possible hyperparameter values.

| Hyperparameter | Possible Values |
|:---:|:---:|
| Solver | 'adam', 'rmsprop', 'adagrad', 'adadelta', 'adamax', 'nadam' |
| Number of epochs | 1, 5, 10, 20, 30, 50, 70, 100, 150, 200 |
| Batch size | 1, 4, 8, 16, 32, 64 |

Hyperparameter tuning was performed with the grid search (GS) procedure. Each of the hyperparameter combinations created using the GS procedure will be individually tested for each of the CNNs. While GS does not guarantee that the best possible hyperparameter combination will be found, it allows a thorough search of hyperparameter values, which are selected based on best practices, as found in previous research [8].

Models generated through the training and testing process with hyperparameters selected using the GS method are evaluated using the Area Under Receiver Operating Characteristic Curve (AUC). AUC is a commonly used measurement of ML method performance, which allows the determination of the classification model quality through the observation of the True Positive Rate ($TPR$) calculated using the number of True Positive ($TP$) and False Negative ($FN$) values [24,25]:

$$TPR = \frac{TP}{TP + FN}.\tag{2}$$

In addition to AUC, several different metrics were used to further validate the model. Based on the $TP$ and $FN$ values, along with their accompanying error types—False Positive ($FP$) and True Negative ($TN$) values, the following metrics were calculated:

- Accuracy $ACC$, calculated as $(TP + TN)/(TP + FP + FN + TN)$;
- Precision $P$, calculated as $TP/(TP + FP)$;
- Recall $R$, calculated as $TP/(TP + FN)$;
- Sensitivity $Sens$, calculated as $TN/(FP + TN)$;
- Specificity $Spec$, calculated as $TN/(FP + TN)$; and

- F1 score, calculated as $2 \cdot TP/(2 \cdot TP + FP + FN)$.

While the AUC was used as the main metric for model tuning, the additional metrics allow us a more detailed look at the exact nature of the errors our models are producing on the data.

To address the problem of model robustness and overfitting, 5-fold cross-validation was applied. This cross-validation is performed by randomly splitting the dataset into 5 equally sized parts. Then, the training and testing procedure is repeated 5 times, with a different dataset part being used as the testing set each time [26]. To clarify this, the training and testing procedures need to be explained. To perform the training, the initial values of filters within the CNNs are set to random values. Then, an image is used as the input to the architecture, resulting in the value of the output neuron between 0 and 1. This output value indicates the probability that the image belongs to a certain class and can be expressed as:

$$\hat{y} = [p_{positive}, p_{negative}], \tag{3}$$

where $p_{negative}$ represents the predicted probability of the sample image belonging to the negative label and $p_{positive}$ indicates the probability of the sample image belonging to the positive label [27]. As the real label of the image is known, due to the images being manually labeled, the error of this prediction can be calculated using the so-called cost function. The cost function indicates the error between the predicted label $\hat{y}$ and the true label $y$. This process is referred to as forward propagation. The error is then propagated back through the CNN architecture in the process called back-propagation, and the internal parameters of the model are adjusted based on the value of the error [28]. This is the training process that adjusts the internal parameter values of the neural network model. The testing process involves the forward propagation of the separate part of the dataset which has not been used in the model training. The forward propagation is repeated for each image, with the model error noted, but no adjustment is performed [29,30]. Based on the noted classification values, the calculation of the model score can be performed, to determine how well the model performs the classification task on the given dataset. The described approach has a few issues. The main issue is overfitting, where the training process can be repeated for too many epochs, causing the model to generalize poorly beyond the data used for modeling. Another issue is the poor data distribution due to randomized selection—such as having a large number of unacceptable images in one of the subsets. As the selection between the training and testing sets is performed randomly, the selected training data may conform extremely well to the trained model, but the general model performance may in reality be extremely poor [31]. The K-fold method aims to address this by using the entire available dataset both for training and testing data [32], as visualized in Figure 3.

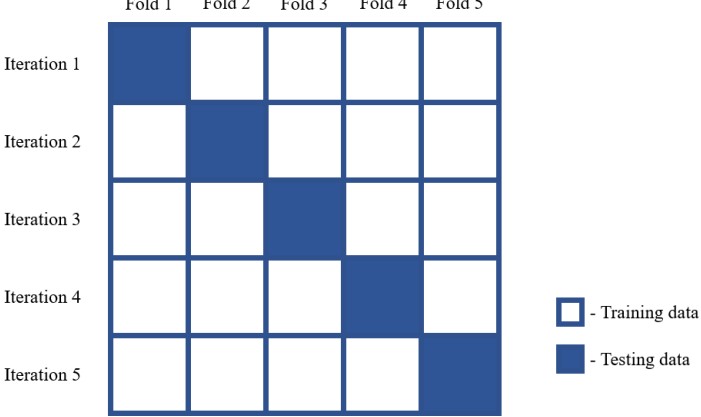

**Figure 3.** A visualization of the 5-fold cross-validation procedure.

After the K-fold process is repeated five times, the result can be expressed as the average value across the data folds, along with the standard error of the model. The average score provides us with information on the model performance—with a large value being preferable, while the standard error indicates the model robustness, where a lower value is better as it indicates a more uniform performance of the classification model across different datasets [26,33]. This ensures that the dataset is split into training and testing data using an 80:20 ratio split. The training dataset is then split further into training and validation sets using a 75:25 ratio (60% of the total data for training, and 20% for internal validation). The test data are not used during the fitting process and are only evaluated using trained models, which yields the presented results. The results are expressed as the average AUC across the five folds ($\overline{AUC}$), and the standard error of the $\overline{AUC}$ folds ($\sigma_{\overline{AUC}}$).

Utilized CNN Architectures

A total of five architectures were used: ResNet50, ResNet101, ResNet152, Xception, and AlexNet. These architectures were selected based on their performance in similar tasks when observing state-of-the-art research. The first considered architecture is Alexnet, which is a relatively simple, 8-layer feed-forward architecture. The benefits of Alexnet are its robustness and computational efficiency—but it often performs more poorly in comparison with more complex networks, especially on higher-resolution images [34,35]. One of the common issues with CNNs is that deeper networks may cause issues with the so-called gradient vanishing [36] This problem can cause the models to be untrainable. ResNet architectures address this by using residual learning blocks, which include a skip association, using the identity activation functions. This allows building significantly deeper networks (50, 101, or even 152 layers), which results in more complex problem-solving capabilities, associated with deeper networks, to be exhibited [37,38]. In addition to minimization of the vanishing gradient problem, these models also offer a fast training process. Three different ResNet configurations were used—ResNet50 [39], ResNet101 [38], and ResNet152 [40], where the number associated with ResNet indicates the number of used residual blocks [41]. Xception network is the last utilized network model. It is based on Depthwise Separable Convolution [42] which is an operation consisting of a depthwise convolution followed by a pointwise convolution [43]. The benefit of this model is high performance even in comparison to similar models such as Inception, but the model is very computationally expensive to train [44,45].

### 2.4. Considered Approaches

Multiple classification approaches have been considered, regarding dataset distribution and hybrid use of the described architectures. For each of the described approaches, a full training procedure was performed, meaning that all five described CNNs have been used, with full GS and cross-validation procedures on each of the separate datasets. In other words, this means that each of the CNN architectures, as presented in the previous section, has been used for each of the separate classifiers presented in the upcoming section. For example, in the approach where the left and right knees are used separately, with the first neural network used to discriminate the orientation and then two networks are used to classify the results as acceptable/unacceptable, each of the aforementioned CNN architectures has been tested for each of the tasks. The first tested approach is the most straightforward, and only concerns taking the LRK dataset and using it as the training/test set directly within the ML methodology as described. This is the most straightforward approach, but the possibility of poor performance due to image orientation differences has to be considered. CNNs are sensitive to image orientation [46], so it was suspected that this may lower the performance when this approach was applied. The schematic view of this approach is provided in Figure 4.

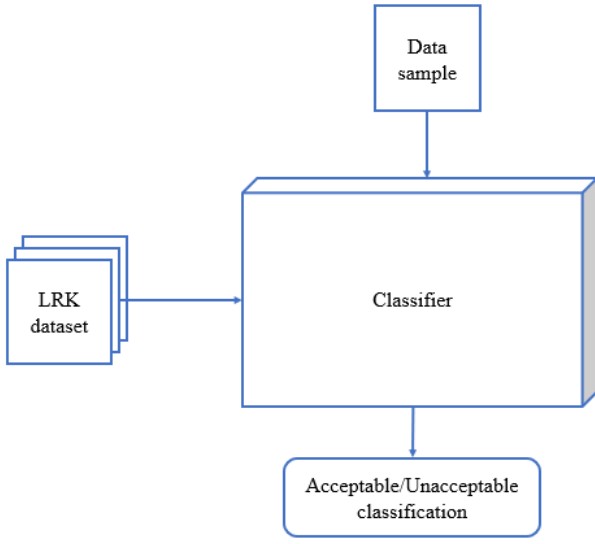

**Figure 4.** LRK approach (LRK—Left+Right Knees).

The second considered approach is a hybrid approach consisting of a network that first determines the orientation of the image, using the OD dataset. The LK and RK datasets are then used to train two separate models. Each of these models classifies the images as "acceptable" and "unacceptable". Due to CNN sensitivity to image orientation, model convergence to a higher-quality solution may be easier when a single orientation of KXE is used. A schematic view of the OD-LK/RK hybrid model is given in Figure 5.

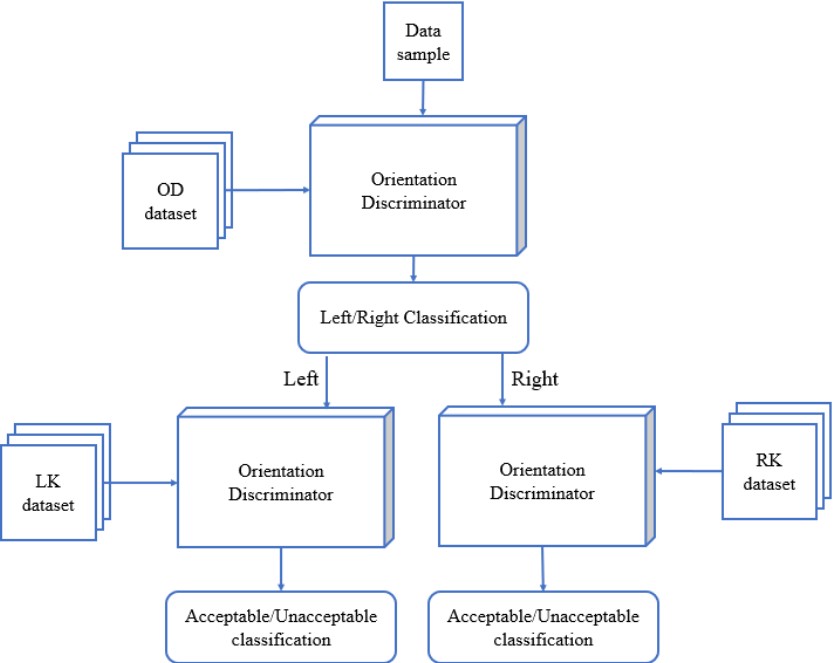

**Figure 5.** OD-LK/RK hybrid approach schematic (OD—Orientation Discrimination, LK—Left Knees, RK—Right Knees).

The last tested approach is the use of the model trained with all images, but flipped in the same orientation. For the testing, the model trained with the OD dataset is used to determine the image orientation. The additional step is added depending on the class of the image determined by the OD, where the image that does not fit the classifier orientation is flipped to match it. Two variants of this hybrid approach have been tested—one with

images being horizontally flipped when the OD detects an RK image and the classifier being trained with an RFK dataset, and another where the image is flipped when the discriminator detects an LK image, and the classifier is trained using LFK dataset. The schematic of this approach is again provided in Figure 6.

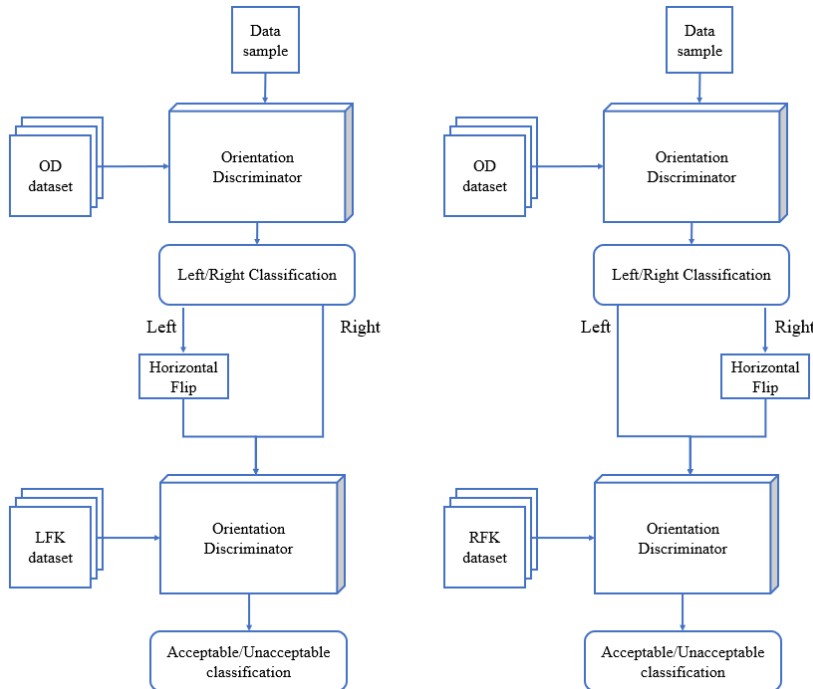

**Figure 6.** OD-LFK/RFK hybrid approach schematic. (OD—Orientation Discrimination, LFK—Left Flipped Knees, RFK—Right Flipped Knees).

All the described networks are implemented using the Tensorflow library (version 2.3.0) and trained using five NVIDIA RTX Quadro 6000 GPUs.

## 3. Results

The results of each approach will be shown separately, with hyperparameters of the best solutions being presented too. In the end, the best overall solution will be presented.

### 3.1. Kappa Index

As noted, kappa index analysis was performed to determine how well the reporting radiographers that labeled the dataset agree with each other. Comparison of the labels between the reporting radiographers yielded a kappa value [47] of 0.75 (87.43%) on a subsample of 200 KXE.

### 3.2. Orientation Discrimination Model Results

As orientation discrimination is used in other approaches, the results of it are shown separately in Table 3.

The best results for the discrimination of image orientation are achieved by the ResNet152 architecture. The results achieved are $\overline{AUC}$ of 0.99, with a low standard error of 0.00, for the input image with the dimensions equal to $224 \times 224 \times 1$. The accuracy and the F1 score of the network are similar. Recall score and sensitivity are both higher than precision and specificity. This holds through for almost all of the metrics and CNNs used for orientation discrimination. Observing the other metrics, it can be noticed that a larger image size tends to lower the standard error while keeping the AUC score nearly the same. According to the results in Table 3, the network that was used as OD for hybrid models was ResNet152 with hyperparameters given in Table 4. This was true for all the image

sizes. The magnitude of standard errors is relatively low compared ith the achieved scores, especially for more complex networks.

**Table 3.** Results of the orientation discrimination model, for each used architecture, and image size, expressed as average metrics (AUC, Accuracy, F1, Recall, Precision, Sensitivity, Specificity) and standard errors.

| | 224 × 224 | | 448 × 448 | | 896 × 896 | |
|---|---|---|---|---|---|---|
| | $\overline{AUC}$ | $\sigma_{\overline{AUC}}$ | $\overline{AUC}$ | $\sigma_{\overline{AUC}}$ | $\overline{AUC}$ | $\sigma_{\overline{AUC}}$ |
| AlexNet | 0.94 | 0.03 | 0.94 | 0.03 | 0.95 | 0.02 |
| ResNet50 | 0.92 | 0.11 | 0.92 | 0.10 | 0.93 | 0.09 |
| ResNet101 | 0.95 | 0.08 | 0.96 | 0.08 | 0.95 | 0.07 |
| ResNet152 | 0.99 | 0.00 | 0.98 | 0.01 | 0.99 | 0.00 |
| Xception | 0.97 | 0.09 | 0.96 | 0.03 | 0.97 | 0.04 |
| | $\overline{ACC}$ | $\sigma_{\overline{ACC}}$ | $\overline{ACC}$ | $\sigma_{\overline{ACC}}$ | $\overline{ACC}$ | $\sigma_{\overline{ACC}}$ |
| AlexNet | 0.94 | 0.04 | 0.94 | 0.04 | 0.95 | 0.03 |
| ResNet50 | 0.92 | 0.10 | 0.93 | 0.09 | 0.94 | 0.06 |
| ResNet101 | 0.95 | 0.06 | 0.96 | 0.05 | 0.95 | 0.05 |
| ResNet152 | 0.99 | 0.01 | 0.98 | 0.02 | 0.99 | 0.01 |
| Xception | 0.97 | 0.08 | 0.96 | 0.04 | 0.96 | 0.04 |
| | $\overline{F1}$ | $\sigma_{\overline{F1}}$ | $\overline{F1}$ | $\sigma_{\overline{F1}}$ | $\overline{F1}$ | $\sigma_{\overline{F1}}$ |
| AlexNet | 0.94 | 0.04 | 0.94 | 0.04 | 0.95 | 0.04 |
| ResNet50 | 0.92 | 0.09 | 0.93 | 0.09 | 0.94 | 0.08 |
| ResNet101 | 0.95 | 0.07 | 0.96 | 0.04 | 0.95 | 0.05 |
| ResNet152 | 0.99 | 0.01 | 0.98 | 0.01 | 0.99 | 0.00 |
| Xception | 0.97 | 0.08 | 0.95 | 0.03 | 0.96 | 0.04 |
| | $\overline{P}$ | $\sigma_{\overline{P}}$ | $\overline{P}$ | $\sigma_{\overline{P}}$ | $\overline{P}$ | $\sigma_{\overline{P}}$ |
| AlexNet | 0.96 | 0.03 | 0.97 | 0.03 | 0.96 | 0.03 |
| ResNet50 | 0.94 | 0.09 | 0.93 | 0.09 | 0.92 | 0.08 |
| ResNet101 | 0.98 | 0.07 | 0.98 | 0.03 | 0.96 | 0.04 |
| ResNet152 | 0.99 | 0.01 | 0.99 | 0.01 | 1,00 | 0.00 |
| Xception | 0.99 | 0.08 | 0.99 | 0.02 | 0.99 | 0.03 |
| | $\overline{R}$ | $\sigma_{\overline{R}}$ | $\overline{R}$ | $\sigma_{\overline{R}}$ | $\overline{R}$ | $\sigma_{\overline{R}}$ |
| AlexNet | 0.93 | 0.05 | 0.92 | 0.06 | 0.94 | 0.05 |
| ResNet50 | 0.91 | 0.10 | 0.93 | 0.09 | 0.96 | 0.08 |
| ResNet101 | 0.93 | 0.07 | 0.94 | 0.06 | 0.95 | 0.07 |
| ResNet152 | 0.99 | 0.01 | 0.97 | 0.02 | 0.99 | 0.00 |
| Xception | 0.96 | 0.08 | 0.92 | 0.04 | 0.94 | 0.04 |
| | $\overline{SPEC}$ | $\sigma_{\overline{SPEC}}$ | $\overline{SPEC}$ | $\sigma_{\overline{SPEC}}$ | $\overline{SPEC}$ | $\sigma_{\overline{SPEC}}$ |
| AlexNet | 0.96 | 0.03 | 0.97 | 0.03 | 0.96 | 0.03 |
| ResNet50 | 0.94 | 0.09 | 0.93 | 0.09 | 0.92 | 0.08 |
| ResNet101 | 0.98 | 0.07 | 0.98 | 0.03 | 0.96 | 0.04 |
| ResNet152 | 0.99 | 0.01 | 0.99 | 0.01 | 1,00 | 0.00 |
| Xception | 0.99 | 0.08 | 0.99 | 0.02 | 0.99 | 0.03 |
| | $\overline{SENS}$ | $\sigma_{\overline{SENS}}$ | $\overline{SENS}$ | $\sigma_{\overline{SENS}}$ | $\overline{SENS}$ | $\sigma_{\overline{SENS}}$ |
| AlexNet | 0.93 | 0.05 | 0.92 | 0.06 | 0.94 | 0.05 |
| ResNet50 | 0.91 | 0.10 | 0.93 | 0.09 | 0.96 | 0.08 |
| ResNet101 | 0.93 | 0.07 | 0.94 | 0.06 | 0.95 | 0.07 |
| ResNet152 | 0.99 | 0.01 | 0.97 | 0.02 | 0.99 | 0.00 |
| Xception | 0.96 | 0.08 | 0.92 | 0.04 | 0.94 | 0.04 |

**Table 4.** Hyperparameters for the highest-performing OD network (ResNet152), used to achieve the scores in Table 3.

| Image Size | Solver | Batch Size | Epochs |
|---|---|---|---|
| 224 × 224 | 'adam' | 64 | 70 |
| 448 × 448 | 'adamax' | 32 | 70 |
| 896 × 896 | 'adam' | 32 | 100 |

### 3.3. LRK Results

The first approach whose results we discuss is the simplest, with all images being used for training without any orientation discrimination. The results are given in Table 5.

**Table 5.** Results of models trained with the LRK approach, for each used architecture and image size, expressed as average metrics (AUC, Accuracy, F1, Recall, Precision, Sensitivity, Specificity) and standard errors.

| | 224 × 224 | | 448 × 448 | | 896 × 896 | |
|---|---|---|---|---|---|---|
| | $\overline{AUC}$ | $\sigma_{\overline{AUC}}$ | $\overline{AUC}$ | $\sigma_{\overline{AUC}}$ | $\overline{AUC}$ | $\sigma_{\overline{AUC}}$ |
| AlexNet | 0.76 | 0.16 | 0.78 | 0.12 | 0.81 | 0.11 |
| ResNet50 | 0.83 | 0.05 | 0.84 | 0.04 | 0.84 | 0.04 |
| ResNet101 | 0.84 | 0.04 | 0.84 | 0.04 | 0.85 | 0.03 |
| ResNet152 | 0.91 | 0.03 | 0.92 | 0.04 | 0.94 | 0.02 |
| Xception | 0.85 | 0.04 | 0.86 | 0.03 | 0.89 | 0.04 |
| | $\overline{ACC}$ | $\sigma_{\overline{ACC}}$ | $\overline{ACC}$ | $\sigma_{\overline{ACC}}$ | $\overline{ACC}$ | $\sigma_{\overline{ACC}}$ |
| AlexNet | 0.76 | 0.21 | 0.81 | 0.17 | 0.82 | 0.12 |
| ResNet50 | 0.83 | 0.16 | 0.84 | 0.14 | 0.84 | 0.12 |
| ResNet101 | 0.84 | 0.16 | 0.84 | 0.15 | 0.85 | 0.12 |
| ResNet152 | 0.91 | 0.12 | 0.92 | 0.11 | 0.94 | 0.09 |
| Xception | 0.85 | 0.16 | 0.86 | 0.13 | 0.90 | 0.11 |
| | $\overline{F1}$ | $\sigma_{\overline{F1}}$ | $\overline{F1}$ | $\sigma_{\overline{F1}}$ | $\overline{F1}$ | $\sigma_{\overline{F1}}$ |
| AlexNet | 0.76 | 0.20 | 0.79 | 0.19 | 0.80 | 0.13 |
| ResNet50 | 0.83 | 0.16 | 0.85 | 0.13 | 0.84 | 0.14 |
| ResNet101 | 0.84 | 0.16 | 0.84 | 0.14 | 0.92 | 0.10 |
| ResNet152 | 0.90 | 0.15 | 0.92 | 0.11 | 0.94 | 0.06 |
| Xception | 0.85 | 0.14 | 0.86 | 0.13 | 0.89 | 0.11 |
| | $\overline{R}$ | $\sigma_{\overline{R}}$ | $\overline{R}$ | $\sigma_{\overline{R}}$ | $\overline{R}$ | $\sigma_{\overline{R}}$ |
| AlexNet | 0.81 | 0.17 | 0.85 | 0.15 | 0.81 | 0.13 |
| ResNet50 | 0.80 | 0.17 | 0.90 | 0.11 | 0.86 | 0.13 |
| ResNet101 | 0.85 | 0.16 | 0.87 | 0.13 | 0.90 | 0.10 |
| ResNet152 | 0.95 | 0.15 | 0.95 | 0.11 | 0.95 | 0.06 |
| Xception | 0.88 | 0.13 | 0.90 | 0.11 | 0.90 | 0.10 |
| | $\overline{P}$ | $\sigma_{\overline{P}}$ | $\overline{P}$ | $\sigma_{\overline{P}}$ | $\overline{P}$ | $\sigma_{\overline{P}}$ |
| AlexNet | 0.72 | 0.24 | 0.74 | 0.26 | 0.80 | 0.12 |
| ResNet50 | 0.86 | 0.15 | 0.80 | 0.15 | 0.82 | 0.15 |
| ResNet101 | 0.83 | 0.17 | 0.82 | 0.16 | 0.95 | 0.09 |
| ResNet152 | 0.86 | 0.15 | 0.90 | 0.11 | 0.94 | 0.07 |
| Xception | 0.82 | 0.16 | 0.82 | 0.15 | 0.88 | 0.12 |
| | $\overline{SENS}$ | $\sigma_{\overline{SENS}}$ | $\overline{SENS}$ | $\sigma_{\overline{SENS}}$ | $\overline{SENS}$ | $\sigma_{\overline{SENS}}$ |
| AlexNet | 0.81 | 0.17 | 0.85 | 0.15 | 0.81 | 0.13 |
| ResNet50 | 0.80 | 0.17 | 0.90 | 0.11 | 0.86 | 0.13 |
| ResNet101 | 0.85 | 0.16 | 0.87 | 0.13 | 0.90 | 0.10 |
| ResNet152 | 0.95 | 0.15 | 0.95 | 0.11 | 0.95 | 0.06 |
| Xception | 0.88 | 0.13 | 0.90 | 0.11 | 0.90 | 0.10 |
| | $\overline{SPEC}$ | $\sigma_{\overline{SPEC}}$ | $\overline{SPEC}$ | $\sigma_{\overline{SPEC}}$ | $\overline{SPEC}$ | $\sigma_{\overline{SPEC}}$ |
| AlexNet | 0.72 | 0.24 | 0.74 | 0.26 | 0.80 | 0.12 |
| ResNet50 | 0.86 | 0.15 | 0.80 | 0.15 | 0.82 | 0.15 |
| ResNet101 | 0.83 | 0.17 | 0.82 | 0.16 | 0.95 | 0.09 |
| ResNet152 | 0.86 | 0.15 | 0.90 | 0.11 | 0.94 | 0.07 |
| Xception | 0.82 | 0.16 | 0.82 | 0.15 | 0.88 | 0.12 |

The data show that the best scores are achieved using the largest tested architecture–ResNet152, which was trained using the adamax solver, with a batch size of 32 for 50 epochs. The results show that a larger-sized input image has some influence on the score, with a significant score increase following the increase in input image size. The magnitude of errors decreases with the complexity of the network suggesting that more complex networks achieve models that show better generalization.

### 3.4. OD-LK/RK Results

The first hybrid approach is using the OD (scores given in Table 3). The scores are given in Table 6, for the entire model, not just the classifier, while including the discriminator.

**Table 6.** Results of models trained with OD-LK/RK approach, for each used architecture and image size, expressed as average metrics (AUC, Accuracy, F1, Recall, Precision, Sensitivity, Specificity) and standard errors.

| | $224 \times 224$ | | $448 \times 448$ | | $896 \times 896$ | |
|---|---|---|---|---|---|---|
| | $\overline{AUC}$ | $\sigma_{\overline{AUC}}$ | $\overline{AUC}$ | $\sigma_{\overline{AUC}}$ | $\overline{AUC}$ | $\sigma_{\overline{AUC}}$ |
| AlexNet | 0.81 | 0.10 | 0.84 | 0.08 | 0.86 | 0.07 |
| ResNet50 | 0.90 | 0.03 | 0.91 | 0.03 | 0.92 | 0.04 |
| ResNet101 | 0.91 | 0.04 | 0.92 | 0.03 | 0.94 | 0.02 |
| ResNet152 | 0.92 | 0.02 | 0.92 | 0.01 | 0.94 | 0.01 |
| Xception | 0.92 | 0.05 | 0.93 | 0.03 | 0.93 | 0.02 |
| | $\overline{ACC}$ | $\sigma_{\overline{ACC}}$ | $\overline{ACC}$ | $\sigma_{\overline{ACC}}$ | $\overline{ACC}$ | $\sigma_{\overline{ACC}}$ |
| AlexNet | 0.81 | 0.17 | 0.84 | 0.10 | 0.86 | 0.12 |
| ResNet50 | 0.90 | 0.10 | 0.91 | 0.09 | 0.92 | 0.07 |
| ResNet101 | 0.92 | 0.08 | 0.92 | 0.08 | 0.94 | 0.06 |
| ResNet152 | 0.92 | 0.09 | 0.92 | 0.08 | 0.94 | 0.06 |
| Xception | 0.92 | 0.09 | 0.93 | 0.08 | 0.93 | 0.07 |
| | $\overline{F1}$ | $\sigma_{\overline{F1}}$ | $\overline{F1}$ | $\sigma_{\overline{F1}}$ | $\overline{F1}$ | $\sigma_{\overline{F1}}$ |
| AlexNet | 0.81 | 0.12 | 0.84 | 0.09 | 0.86 | 0.09 |
| ResNet50 | 0.90 | 0.10 | 0.90 | 0.09 | 0.92 | 0.07 |
| ResNet101 | 0.92 | 0.09 | 0.92 | 0.08 | 0.94 | 0.06 |
| ResNet152 | 0.92 | 0.09 | 0.92 | 0.08 | 0.94 | 0.06 |
| Xception | 0.92 | 0.09 | 0.93 | 0.08 | 0.93 | 0.07 |
| | $\overline{R}$ | $\sigma_{\overline{R}}$ | $\overline{R}$ | $\sigma_{\overline{R}}$ | $\overline{R}$ | $\sigma_{\overline{R}}$ |
| AlexNet | 0.82 | 0.12 | 0.86 | 0.09 | 0.88 | 0.10 |
| ResNet50 | 0.91 | 0.09 | 0.88 | 0.10 | 0.95 | 0.06 |
| ResNet101 | 0.93 | 0.10 | 0.92 | 0.08 | 0.96 | 0.04 |
| ResNet152 | 0.94 | 0.08 | 0.93 | 0.07 | 0.95 | 0.05 |
| Xception | 0.93 | 0.09 | 0.95 | 0.08 | 0.93 | 0.07 |
| | $\overline{P}$ | $\sigma_{\overline{P}}$ | $\overline{P}$ | $\sigma_{\overline{P}}$ | $\overline{P}$ | $\sigma_{\overline{P}}$ |
| AlexNet | 0.80 | 0.12 | 0.82 | 0.08 | 0.84 | 0.09 |
| ResNet50 | 0.90 | 0.10 | 0.92 | 0.08 | 0.90 | 0.09 |
| ResNet101 | 0.92 | 0.09 | 0.92 | 0.08 | 0.92 | 0.08 |
| ResNet152 | 0.91 | 0.10 | 0.91 | 0.08 | 0.93 | 0.07 |
| Xception | 0.92 | 0.09 | 0.92 | 0.08 | 0.93 | 0.07 |
| | $\overline{SENS}$ | $\sigma_{\overline{SENS}}$ | $\overline{SENS}$ | $\sigma_{\overline{SENS}}$ | $\overline{SENS}$ | $\sigma_{\overline{SENS}}$ |
| AlexNet | 0.82 | 0.12 | 0.86 | 0.09 | 0.88 | 0.10 |
| ResNet50 | 0.91 | 0.09 | 0.88 | 0.10 | 0.95 | 0.06 |
| ResNet101 | 0.93 | 0.10 | 0.92 | 0.08 | 0.96 | 0.04 |
| ResNet152 | 0.94 | 0.08 | 0.93 | 0.07 | 0.95 | 0.05 |
| Xception | 0.93 | 0.09 | 0.95 | 0.08 | 0.93 | 0.07 |
| | $\overline{SPEC}$ | $\sigma_{\overline{SPEC}}$ | $\overline{SPEC}$ | $\sigma_{\overline{SPEC}}$ | $\overline{SPEC}$ | $\sigma_{\overline{SPEC}}$ |
| AlexNet | 0.80 | 0.12 | 0.82 | 0.08 | 0.84 | 0.09 |
| ResNet50 | 0.90 | 0.10 | 0.92 | 0.08 | 0.90 | 0.09 |
| ResNet101 | 0.92 | 0.09 | 0.92 | 0.08 | 0.92 | 0.08 |
| ResNet152 | 0.91 | 0.10 | 0.91 | 0.08 | 0.93 | 0.07 |
| Xception | 0.92 | 0.09 | 0.92 | 0.08 | 0.93 | 0.07 |

The results show that ResNet152 again achieves the best results. In this case, ResNet152 used an adamax solver, as in the previously discussed approach, but with a lower batch size of 16 when trained for 50 epochs. The increase in classification performance is again seen when the input image size is increased, and the best-performing model is again achieved when using the $896 \times 896$ image size.

### 3.5. OD-LFK and OD-LRK Results

Finally, the last hybrid approach is discussed. The results are again given for the whole system, including the discriminator. Both laterality approaches were tested—using only left or right-oriented images. The results for the models which use only left-oriented images (right-oriented images have been horizontally flipped) are given in Table 7, while the results for the configuration which uses only right-oriented images (left-oriented images flipped) are presented in Table 8.

**Table 7.** Results of model training for OD-LFK approach, for each used architecture and image size, expressed as average metrics (AUC, Accuracy, F1, Recall, Precision, Sensitivity, Specificity) and standard errors.

| | 224 × 224 | | 448 × 448 | | 896 × 896 | |
|---|---|---|---|---|---|---|
| | $\overline{AUC}$ | $\sigma_{\overline{AUC}}$ | $\overline{AUC}$ | $\sigma_{\overline{AUC}}$ | $\overline{AUC}$ | $\sigma_{\overline{AUC}}$ |
| AlexNet | 0.82 | 0.06 | 0.83 | 0.05 | 0.83 | 0.06 |
| ResNet50 | 0.91 | 0.02 | 0.92 | 0.01 | 0.91 | 0.01 |
| ResNet101 | 0.92 | 0.03 | 0.92 | 0.01 | 0.93 | 0.01 |
| ResNet152 | 0.94 | 0.02 | 0.95 | 0.01 | 0.95 | 0.01 |
| Xception | 0.96 | 0.02 | 0.95 | 0.01 | 0.96 | 0.01 |
| | $\overline{ACC}$ | $\sigma_{\overline{ACC}}$ | $\overline{ACC}$ | $\sigma_{\overline{ACC}}$ | $\overline{ACC}$ | $\sigma_{\overline{ACC}}$ |
| AlexNet | 0.83 | 0.11 | 0.83 | 0.12 | 0.83 | 0.09 |
| ResNet50 | 0.91 | 0.07 | 0.92 | 0.08 | 0.91 | 0.08 |
| ResNet101 | 0.92 | 0.08 | 0.92 | 0.06 | 0.95 | 0.06 |
| ResNet152 | 0.95 | 0.05 | 0.95 | 0.05 | 0.96 | 0.06 |
| Xception | 0.96 | 0.04 | 0.95 | 0.05 | 0.96 | 0.05 |
| | $\overline{F1}$ | $\sigma_{\overline{F1}}$ | $\overline{F1}$ | $\sigma_{\overline{F1}}$ | $\overline{F1}$ | $\sigma_{\overline{F1}}$ |
| AlexNet | 0.82 | 0.09 | 0.82 | 0.10 | 0.83 | 0.10 |
| ResNet50 | 0.92 | 0.08 | 0.92 | 0.07 | 0.91 | 0.08 |
| ResNet101 | 0.92 | 0.08 | 0.92 | 0.06 | 0.93 | 0.07 |
| ResNet152 | 0.94 | 0.06 | 0.95 | 0.05 | 0.95 | 0.06 |
| Xception | 0.96 | 0.05 | 0.95 | 0.05 | 0.96 | 0.06 |
| | $\overline{R}$ | $\sigma_{\overline{R}}$ | $\overline{R}$ | $\sigma_{\overline{R}}$ | $\overline{R}$ | $\sigma_{\overline{R}}$ |
| AlexNet | 0.80 | 0.07 | 0.81 | 0.09 | 0.80 | 0.14 |
| ResNet50 | 0.93 | 0.07 | 0.93 | 0.06 | 0.92 | 0.08 |
| ResNet101 | 0.92 | 0.07 | 0.92 | 0.07 | 0.93 | 0.06 |
| ResNet152 | 0.96 | 0.05 | 0.96 | 0.05 | 0.97 | 0.05 |
| Xception | 0.97 | 0.04 | 0.96 | 0.05 | 0.97 | 0.05 |
| | $\overline{P}$ | $\sigma_{\overline{P}}$ | $\overline{P}$ | $\sigma_{\overline{P}}$ | $\overline{P}$ | $\sigma_{\overline{P}}$ |
| AlexNet | 0.85 | 0.12 | 0.84 | 0.12 | 0.87 | 0.07 |
| ResNet50 | 0.91 | 0.09 | 0.92 | 0.08 | 0.90 | 0.09 |
| ResNet101 | 0.92 | 0.08 | 0.92 | 0.06 | 0.93 | 0.07 |
| ResNet152 | 0.92 | 0.08 | 0.94 | 0.06 | 0.94 | 0.06 |
| Xception | 0.95 | 0.06 | 0.95 | 0.05 | 0.95 | 0.06 |
| | $\overline{SENS}$ | $\sigma_{\overline{SENS}}$ | $\overline{SENS}$ | $\sigma_{\overline{SENS}}$ | $\overline{SENS}$ | $\sigma_{\overline{SENS}}$ |
| AlexNet | 0.80 | 0.07 | 0.81 | 0.09 | 0.80 | 0.14 |
| ResNet50 | 0.93 | 0.07 | 0.93 | 0.06 | 0.92 | 0.08 |
| ResNet101 | 0.92 | 0.07 | 0.92 | 0.07 | 0.93 | 0.06 |
| ResNet152 | 0.96 | 0.05 | 0.96 | 0.05 | 0.97 | 0.05 |
| Xception | 0.97 | 0.04 | 0.96 | 0.05 | 0.97 | 0.05 |
| | $\overline{SPEC}$ | $\sigma_{\overline{SPEC}}$ | $\overline{SPEC}$ | $\sigma_{\overline{SPEC}}$ | $\overline{SPEC}$ | $\sigma_{\overline{SPEC}}$ |
| AlexNet | 0.85 | 0.12 | 0.84 | 0.12 | 0.87 | 0.07 |
| ResNet50 | 0.91 | 0.09 | 0.92 | 0.08 | 0.90 | 0.09 |
| ResNet101 | 0.92 | 0.08 | 0.92 | 0.06 | 0.93 | 0.07 |
| ResNet152 | 0.92 | 0.08 | 0.94 | 0.06 | 0.94 | 0.06 |
| Xception | 0.95 | 0.06 | 0.95 | 0.05 | 0.95 | 0.06 |

The results in Table 7 show that the best results are achieved using Xception. Interestingly, the input image size seems to have a significantly smaller influence on the scores,

with the model trained with lower-size images slightly outscoring the models trained with larger ones. Xception achieved the $\overline{AUC}$ score of 0.96 with a standard error of 0.01 across folds, which is a very low error magnitude, suggesting very stable models. Higher variance is present when observing the other metrics, but the average scores of the metrics are still high. The model in question was trained for 100 epochs, using the batch size of 8 and the nadam solver.

**Table 8.** Results of model training for OD-RFK approach, for each used architecture and image size, expressed as average metrics (AUC, Accuracy, F1, Recall, Precision, Sensitivity, Specificity) and standard errors.

| | 224 × 224 | | 448 × 448 | | 896 × 896 | |
|---|---|---|---|---|---|---|
| | $\overline{AUC}$ | $\sigma_{\overline{AUC}}$ | $\overline{AUC}$ | $\sigma_{\overline{AUC}}$ | $\overline{AUC}$ | $\sigma_{\overline{AUC}}$ |
| AlexNet | 0.83 | 0.06 | 0.83 | 0.05 | 0.82 | 0.06 |
| ResNet50 | 0.92 | 0.01 | 0.92 | 0.02 | 0.93 | 0.01 |
| ResNet101 | 0.92 | 0.02 | 0.93 | 0.01 | 0.92 | 0.01 |
| ResNet152 | 0.84 | 0.01 | 0.94 | 0.01 | 0.95 | 0.01 |
| Xception | 0.97 | 0.01 | 0.97 | 0.02 | 0.97 | 0.01 |
| | $\overline{ACC}$ | $\sigma_{\overline{ACC}}$ | $\overline{ACC}$ | $\sigma_{\overline{ACC}}$ | $\overline{ACC}$ | $\sigma_{\overline{ACC}}$ |
| AlexNet | 0.85 | 0.12 | 0.83 | 0.09 | 0.82 | 0.10 |
| ResNet50 | 0.93 | 0.07 | 0.92 | 0.06 | 0.93 | 0.06 |
| ResNet101 | 0.92 | 0.08 | 0.93 | 0.05 | 0.92 | 0.07 |
| ResNet152 | 0.84 | 0.13 | 0.94 | 0.05 | 0.95 | 0.06 |
| Xception | 0.97 | 0.05 | 0.97 | 0.04 | 0.97 | 0.03 |
| | $\overline{F1}$ | $\sigma_{\overline{F1}}$ | $\overline{F1}$ | $\sigma_{\overline{F1}}$ | $\overline{F1}$ | $\sigma_{\overline{F1}}$ |
| AlexNet | 0.83 | 0.11 | 0.83 | 0.14 | 0.82 | 0.09 |
| ResNet50 | 0.92 | 0.07 | 0.92 | 0.06 | 0.93 | 0.06 |
| ResNet101 | 0.92 | 0.07 | 0.94 | 0.05 | 0.92 | 0.07 |
| ResNet152 | 0.84 | 0.13 | 0.95 | 0.05 | 0.95 | 0.06 |
| Xception | 0.97 | 0.05 | 0.97 | 0.05 | 0.97 | 0.03 |
| | $\overline{R}$ | $\sigma_{\overline{R}}$ | $\overline{R}$ | $\sigma_{\overline{R}}$ | $\overline{R}$ | $\sigma_{\overline{R}}$ |
| AlexNet | 0.81 | 0.15 | 0.80 | 0.14 | 0.81 | 0.13 |
| ResNet50 | 0.93 | 0.06 | 0.93 | 0.06 | 0.95 | 0.06 |
| ResNet101 | 0.92 | 0.07 | 0.95 | 0.05 | 0.94 | 0.06 |
| ResNet152 | 0.84 | 0.12 | 0.95 | 0.05 | 0.94 | 0.06 |
| Xception | 0.99 | 0.04 | 0.98 | 0.05 | 0.99 | 0.02 |
| | $\overline{P}$ | $\sigma_{\overline{P}}$ | $\overline{P}$ | $\sigma_{\overline{P}}$ | $\overline{P}$ | $\sigma_{\overline{P}}$ |
| AlexNet | 0.86 | 0.08 | 0.86 | 0.13 | 0.83 | 0.07 |
| ResNet50 | 0.91 | 0.07 | 0.92 | 0.05 | 0.92 | 0.06 |
| ResNet101 | 0.92 | 0.08 | 0.94 | 0.06 | 0.91 | 0.08 |
| ResNet152 | 0.84 | 0.14 | 0.95 | 0.05 | 0.96 | 0.05 |
| Xception | 0.95 | 0.06 | 0.96 | 0.05 | 0.95 | 0.05 |
| | $\overline{SENS}$ | $\sigma_{\overline{SENS}}$ | $\overline{SENS}$ | $\sigma_{\overline{SENS}}$ | $\overline{SENS}$ | $\sigma_{\overline{SENS}}$ |
| AlexNet | 0.81 | 0.15 | 0.80 | 0.14 | 0.81 | 0.13 |
| ResNet50 | 0.93 | 0.06 | 0.93 | 0.06 | 0.95 | 0.06 |
| ResNet101 | 0.92 | 0.07 | 0.95 | 0.05 | 0.94 | 0.06 |
| ResNet152 | 0.84 | 0.12 | 0.95 | 0.05 | 0.94 | 0.06 |
| Xception | 0.99 | 0.04 | 0.98 | 0.05 | 0.99 | 0.02 |
| | $\overline{SPEC}$ | $\sigma_{\overline{SPEC}}$ | $\overline{SPEC}$ | $\sigma_{\overline{SPEC}}$ | $\overline{SPEC}$ | $\sigma_{\overline{SPEC}}$ |
| AlexNet | 0.86 | 0.08 | 0.86 | 0.13 | 0.83 | 0.07 |
| ResNet50 | 0.91 | 0.07 | 0.92 | 0.05 | 0.92 | 0.06 |
| ResNet101 | 0.92 | 0.08 | 0.94 | 0.06 | 0.91 | 0.08 |
| ResNet152 | 0.84 | 0.14 | 0.95 | 0.05 | 0.96 | 0.05 |
| Xception | 0.95 | 0.06 | 0.96 | 0.05 | 0.95 | 0.05 |

As seen in Table 8, the best overall results are achieved using Xception trained with the OD-RFK approach. This approach achieves a total average AUC across folds of 0.97, with a standard error of 0.01. All other scores show this network as high-performing, but notably, the model shows higher recall and sensitivity compared with precision and specificity. All networks except the simplest one used–AlexNet, show an extremely low magnitude of standard error across folds. The Xception architecture was trained for 100 epochs using the nadam solver, and a batch size of 4. The low batch size indicates that some regularization was necessary to achieve the best results, as smaller batch sizes can introduce such an effect. The results were not highly influenced by input image size.

*3.6. Best Result Analysis*

As shown in the previous section the best results are achieved using the OD-LRK hybrid approach with ResNet152 as the OD and Xception as the image quality classifier. Due to the nearly identical performance regarding image sizes, $224 \times 224$ and $896 \times 896$ based on average AUC score and standard error both being equal to the second significant decimal, the model selected as the best solution was $224 \times 224$. This was chosen for a few reasons, which are mainly related to the benefits of the smaller image size being used, such images are simpler to store, with smaller generated models and marginally faster execution times. Moreover, smaller image sizes also influence training times. This means that if further model training is performed with newly collected data, those images can be more easily transferred, resulting in faster training.

Figure 7 shows that the average AUC score over folds steadily grows with more epochs, while the standard error steadily falls, until reaching 100 epochs. After 100 epochs, the score across folds starts to decrease, along with a standard error increase. This behavior indicates that the model overfits at higher epochs. To further illustrate the performance of the model as described, a confusion matrix of the entire hybrid approach is shown in Figure 8. A single cross-validation fold is randomly selected and used for evaluation, for brevity. It can be seen that on the selected validation set of 800 images (20%), the selected model classifies "acceptable" images as "unacceptable" and 16 "unacceptable" images as "acceptable".

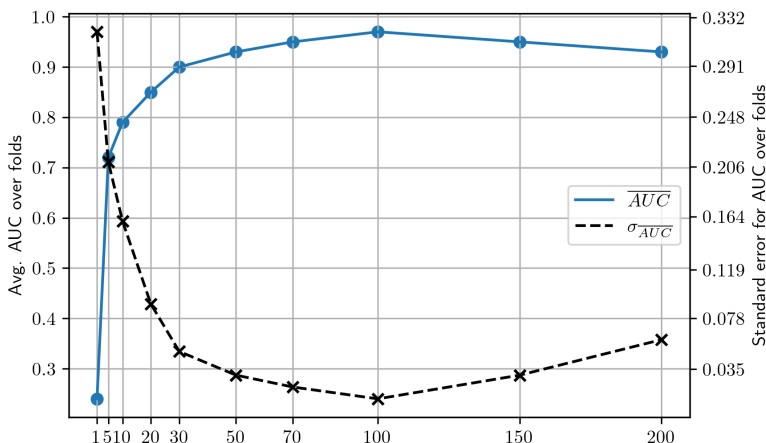

**Figure 7.** The classification performance of the best model across epochs used in training.

Another value worth observing was classification speed (CS). CS can be measured by loading the models into the computer's random-access memory and measuring it, using the Python time library [48]. After repeating this process 100 times, it was noticed that the average CS was $\bar{t}_{wall} = 2.4$ s, $\sigma_{\bar{t}_{wall}} = 0.07$ s. This time was achieved using a workstation with Intel i5-9400 CPU@2.9GHz and 8GB of RAM, indicating that the generated model was fast enough to be used in practice.

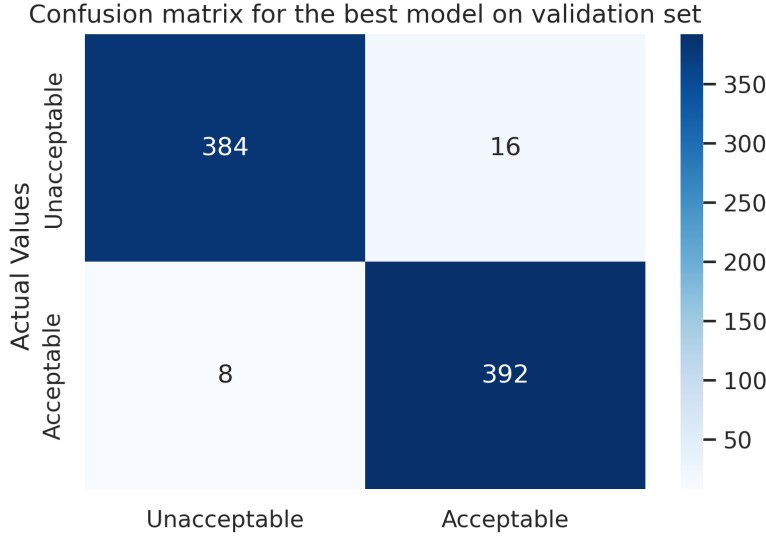

**Figure 8.** Confusion matrix of the best-performing model.

## 4. Discussion

The presented results show certain regularities which will be discussed in this section. Limitations of the study will also be addressed. All of the models have been evaluated using multiple metrics. The main evaluation metric, used in the training procedure, was AUC, with it and Accuracy, F1, Recall, Precision, Sensitivity, and Specificity used for model evaluation on five-fold cross-validation sets. The results show very low variation in average metric scores between certain metric types, such as AUC, F1, and Accuracy—but the standard variations are somewhat higher with accuracy and F1, compared with AUC—indicating that AUC is not sufficient as a sole metric to be used for evaluation. This is probably caused by the fact that the models were initially trained using AUC as a main metric, causing a somewhat better average performance across folds. The same can be seen in the case of precision-specificity and recall-sensitivity metric pairs, which are equal when rounded to two significant decimals. It must be noted that part of the reason for this is a relatively small binary test dataset of only 400 images. Most of the models show higher recall and sensitivity, compared with precision and specificity, which is important to note considering the goal of the system. As this is prevalent across most of the model results, the indication is that this is caused by the images contained in the dataset. Curiously, similar behavior is visible for papers reviewed as part of the related research. Image size does not have a large influence on the model performance, with scores for the same network/approach combination being very similar across all image sizes. Here, the indication is that smaller-sized images can be used for further development of this and similar systems, due to the simpler handling of such data and faster/less computationally expensive model training on smaller image sizes. The first thing to note is the general results of the individual CNN architectures without observing the different approaches. Results show that in most cases, more complex networks generate better results, e.g., ResNet152 shows better results than ResNet50, and Xception shows better results than AlexNet. This is not surprising, due to larger models allowing for a larger number of trainable parameters, which means that they can be trained to address more complex problems. Still, this is not always the case, which indicates the possibility of the larger models failing to be trained with the data present in the respective fold's training/validation sets. When the results of different approaches are observed, they show that there is a definite score benefit to the use of a hybrid model which combines the OD CNN with a Classifier CNN. This indicates multiple things. First, due to OD flipping the images improving the results, it can be assumed that the orientation of the XE does play a factor in the performance of the classifier. In other words, Classifier

networks show better performance in cases where all of the input images are oriented in the same direction. The OD-LFK/RFK approach seems to show a slightly better top performance than OD-LK/RK approach, with top scores around 0.97 when observing AUC for OD-LFK/RFK compared with 0.94 for OD-LK/RK. Within the OD-LFK/RFK approach, OD-RFK has slightly superior results, but these fall within the range of standard error on a five-fold cross-validated dataset, meaning their performance is essentially interchangeable.

*Limitations*

Limitations of the presented study follow the common limitations present in most AI studies. The developed models could perform poorly when applied to certain edge cases which were not contained in the utilized dataset. Different XE acquisition could result in poorer performance if the models are applied without adjustment (e.g., training with transfer learning). Finally, models developed will only perform well on XE of knees, as that is what the model training was performed with—and will probably suffer significant drops in performance if they are attempted to be applied on different XE [2,27]. As the input datasets were standing, weight-bearing KXEs for the diagnosis of osteoarthritis, performance may be different for supine KXE, e.g., for fracture detection, where diagnostic criteria are different and indirect soft tissue findings are as important as the bone itself. Furthermore, all KXEs in the study were acquired using a support stand and an aluminum calibration disk (see Figure 1), which is not used in fracture KXEs. It can be speculated that these features might influence model training, but this remains an area for further study.

## 5. Conclusions

The manuscript presents an automated system for the classification of lateral KXE quality in the acquired dataset. Three separate approaches have been tested, to compare the performance of different combinations of CNNs. When evaluated using different metrics, it can be seen that the best performance is achieved when the ResNet152 is used to discriminate the orientation of XE between left and right, achieving scores of $\overline{AUC} = 0.97 \pm 0.09; \overline{ACC} = 0.97 \pm 0.08; \overline{F1} = 0.97 \pm 0.08; \overline{R} = 0.99 \pm 0.08; \overline{P} = 0.96 \pm 0.06; \overline{Sens} = 0.99 \pm 0.04; \overline{Spec} = 0.95 \pm 0.06$, in combination with an Xception CNN as a classifier that achieved scores of $\overline{AUC} = 0.97 \pm 0.01; \overline{ACC} = 0.97 \pm 0.05; \overline{F1} = 0.97 \pm 0.05; \overline{R} = 0.99 \pm 0.04; \overline{P} = 0.95 \pm 0.06; \overline{Sens} = 0.99 \pm 0.04; \overline{Spec} = 0.95 \pm 0.06$. These results are achieved when combining the networks into an RFK approach—the discriminator detects the orientation of the XE, left-oriented images are flipped to match the right image orientation, and then the classifier is applied. Due to the performance improvements when all images used for training/test/validation are flipped to the same orientation, it can be concluded that such an approach may be beneficial in application to similar problems. Comparing the results presented in this study, it is visible that the achieved results are comparable, if not better, than similar approaches performed on different data. Higher sensitivity than specificity shown in presented scores is consistent with previous related studies.

Our research also concludes that there is a high degree of scale invariance, with larger input images demonstrating small improvements, resulting in longer training times and memory use. This is possibly caused by the fact that the CNNs used are generally developed for smaller images, and indicates that the use of smaller images for similar studies may improve training speed without sacrificing accuracy. The application of the proposed models could greatly assist in the differentiation between XE of acceptable or unacceptable quality. This will contribute to simpler and quicker processing of patients who require XE to be performed before further diagnostic steps. As mentioned, the study has certain limitations. For example, there is no way to determine the performance of different XE collection methodologies, as a single source of images was used. Further research on a wider dataset is needed to assess the performance of different XE acquisition methodologies for other diagnostic procedures, such as non-weight-bearing XEs and ones without an aluminum calibration disk present. As seen with the limitations of the study, there are a number of items to be focused on in future work. Testing and validating the

developed methodology on differently collected XE datasets, especially from multiple hospital sources, is key to ensuring the performance of the developed system in wider use, with the presented work used as a pilot study. Beyond that, the development of similar models for different anatomical regions could serve as a useful study. Finally, further application of AI-based methods on collected images to automate the system further, such as automatic calculation of adjustment angle, could be beneficial, as it would provide more information to XE technicians beyond informing them whether the XE performed is satisfactory or not.

**Author Contributions:** Conceptualization, S.L., S.H., R.A. and Z.C.; methodology: S.L., S.H., R.A., M.W.K. and Z.C.; software, S.B.Š.; validation: S.L.; formal analysis, Z.C.; investigation, S.L. and S.B.Š.; resources, S.L., S.H., R.A., M.W.K. and Z.C.; data curation, S.L., S.H., R.A. and M.W.K.; writing–original draft preparation: S.L. and S.B.Š.; writing–review and editing, S.H., R.A., M.W.K. and Z.C.; visualization: S.B.Š.; project administration, S.L. and Z.C.; funding acquisition: Z.C. and S.B.Š. All authors have read and agreed to the published version of the manuscript.

**Funding:** This research received no external funding.

**Institutional Review Board Statement:** The National Committee on Health Research Ethics (NVK) approved this study and granted access to the patients' X-ray images (Case number: 2012874).

**Informed Consent Statement:** Not applicable.

**Data Availability Statement:** Data used in this study is not made publicly available due to the patient privacy.

**Acknowledgments:** This research has been (partly) supported by the CEEPUS network CIII-HR-0108, European Regional Development Fund under the grant KK.01.1.1.01.0009 (DATACROSS), project CEKOM under the grant KK.01.2.2.03.0004, Erasmus+ project WICT under the grant 2021-1-HR01-KA220-HED-000031177 and University of Rijeka scientific grant uniri-tehnic-18-275-1447.

**Conflicts of Interest:** The authors declare no conflict of interest.

## Abbreviations

The following abbreviations are used in this manuscript:

| | |
|---|---|
| XE | X-ray examinations |
| CT | Computation time |
| OD | Orientation discriminator dataset/model |
| LK | Left knee dataset/model |
| RK | Right knee dataset/model |
| LFK | Left knee with right knee flipped dataset/model |
| RFK | Right knee with left knee flipped dataset/model |
| OD-LK/RK | Model for classification of images with flipped orientation |
| OD-LFK | Orientation discriminator combined with left knee with right knee flipped model |
| OD-RFK | Orientation discriminator combined with right knee with left knee flipped model |

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
