# Peer review of "Quality Assessment Assistance of Lateral Knee X-rays: A Hybrid Convolutional Neural Network Approach"

_mathematics, doi:10.3390/math11102392_

Round 1

Reviewer 1 Report

What is your theoretical contribution to the proposed model? The added value and the differences between this proposal and those that already exist in the literature are not clear.

Minor editing of English language required

Author Response

We would like to thank the reviewer for their comments on our manuscript. Please find our answer to the posed question below. We have marked the changes in text made due to this Reviewer’s comments with blue text.

  • What is your theoretical contribution to the proposed model? The added value and the differences between this proposal and those that already exist in the literature are not clear.

The following explanation has been added to the end of Introduction section:

“The contributions of the presented work can be presented as: 

  • Develop an AI-based system for classification of lateral KXE quality, in order to inform the technician performing the examination immediately if the positioning of the KXE is not satisfactory, 
  • Test approaches combining different CNN architectures in various hybrid setups and compare the performance to standard, single-CNN approach in order to determine the need for the use of advanced CNN approaches, and 
  • Determine which parameters of the entire system generate the best performance, in regards to CNN architecture, hybrid combination of the CNNs, hyperparameters, and image size in order to enable simpler future development of models focused on similar issues”

The knowledge gap compared to the existing work has also been discussed:

“There are a few knowledge gaps that can be identified observing the related work. First, there is a clear lack of application on the specific issue of KXE quality assessment. Second, only a few studies consider the application of existing, tried and tested, CNN architectures and opt for a customized CNN approach. This approach complicates the studies through the introduction of unknown variables, as well as making them harder to reproduce, while they may not provide a benefit. Most of the research does not cross-validate the data, and uses the standard train-test approach. Most authors use images as-is, without attempting to identify if there are performance benefits from using a different size images. It is also important to note the numbers of images used, which range wildly - from less than a hundred images to tens of thousands, with the dataset used in the presented study falling around the middle at 4,000 images. Finally, little testing is done with combinations of various different ML-architectures - despite research in other area showing that these types of methods can yield significantly improved results, only a single paper identified in the related work review applies them for quality assessment [20].”

We hope the reviewer will consider the manuscript significantly improved now that their comments have been addressed

Kindest regards,
the authors.

Reviewer 2 Report

Reviewer's Comments:

In this paper the authors have proposed a system for automatic quality classification of lateral knee X-ray examinations based on convolutional neural networks (CNN). This is an empirical comparison of existing CNNs for lateral knee x-ray classification.

The following points need to be addressed for improving the quality of the manuscript.

Major concern:

·         In introduction section the authors need to specify separately two paragraphs specifying the motivation and contribution of their work. It will be better to list the contribution in two or three main points.

·         The Related work is to be organized in the form of a tabular form. It will be better to provide the strength and weakness of the proposed approach. It should also include theoretical comparison to other approaches.

·         It will be setter to specify the outcomes related to each combination of hyper parametres to specify the best suitable one.

·         Perform a sensitivity analysis of the model.

·         Specify the model outcome with respect to some other classification criteria such as precision, recall, accuracy, f1-score and Gmean.

·         Mention a separate paragraph including the limitation and future work of your study in the conclusion section.

Minor concern:

·         In Line 233 , Table ?? will be Table 3.

Author Response

We would like to thank the reviewer for their review of our manuscript. We have answered the questions point-by-point below. We have marked the changes in text made due to this Reviewer’s comments with green text.

  • In introduction section the authors need to specify separately two paragraphs specifying the motivation and contribution of their work. It will be better to list the contribution in two or three main points.

The motivation for further clarified by adding the following paragraph to the Introduction section:

“The main motivation of the presented work is to simplify the process of KXE examinations for both the patients and the medical staff. The low quality KXEs refer to those KXEs in which the patient's knee is not positioned properly or is misoriented, not allowing for the medical examiner to diagnose the injury properly. When this happens, a new XE must be performed, which puts undue stress on the hospital system as well as lowering the quality of healthcare obtained by the patient, due to the delay introduced before a diagnosis can be made.”

The contributions have been clarified by adding the following text to the end of the Introduction section:

“The contributions of the presented work can be presented as: 

  • Develop an AI-based system for classification of lateral KXE quality, in order to inform the technician performing the examination immediately if the positioning of the KXE is not satisfactory, 
  • Test approaches combining different CNN architectures in various hybrid setups and compare the performance to standard, single-CNN approach in order to determine the need for the use of advanced CNN approaches, and 
  • Determine which parameters of the entire system generate the best performance, in regards to CNN architecture, hybrid combination of the CNNs, hyperparameters, and image size in order to enable simpler future development of models focused on similar issues”
  • The Related work is to be organized in the form of a tabular form. It will be better to provide the strength and weakness of the proposed approach. It should also include theoretical comparison to other approaches.

“Related work” was added as a subsection to the “Introduction” section, describing the related recently published works:

“Zhang et al. (2021) [13] apply the Feature Extraction CNN on the problem of fetal sonograph quality assessment. On the balanced dataset of 14,700 images, authors manage to achieve AUC and F1 scores of 0.96. Lauermann et al. (2019) [14] described the use of a customized deep learning algorithm, based on CNNs, for the problem of optical coherence tomography angiography quality assessment. The models are evaluated using accuracy and achieve the score of 0.97 on the validation dataset. Jalaboi et al. (2023) [15] focus on the application of a custom CNN architecture "ImageQX". The architecture is trained on 36,509 images, out of which the validation set consists of 9,874 photographs. With the architecture at hand, athors achieve the score of 0.73 ± 0.01. Quality assessment on eye fundus photography is performed by Karlsson et al. (2021) [16 ]. They apply a random forest model on features extracted with Fourier transformation on a publicly available DRIMDB dataset, consisting of 216 images. The authors demonstrate the accuracy score of 0.981, sensitivity score of 0.993 and specificity score of 0.958. Mairhofer et al. (2021) [17]. Authors extract the region of interest and classify the quality into one of three classes using a customized CNN based framework. When evaluating on the set of 950 images, authors achieve the accuracy score of 0.94 with their proposed three-step framework. Chabert et al. (2021) [ 18 ] apply multiple methods (linear discriminant analysis, quadratic linear analysis, support vector machine, logistic regression and multilayer perceptron) for assessment of Lumbar MRI quality. Best results are achieved with a combination of methods, with the recall of 0.82 and AUC of 0.97. Coyner et al. (2019) [ 19] demonstrate the use of a custom deep CNNs for quality assessment of retinopathy images. Using a dataset of 4,000 images authors achieve an AUC of 0.97, with the sensitivity of 0.94 and specificity of 0.84. Czajowska et al. (2022) [ 20 ] apply a hybrid system consisting of a deep CNN and a fuzzy reasoning system on quality assessment of author collected dataset of high-frequency facial ultrasounds. The achieved results show that the proposed system achieves a classification accuracy of 0.92. The summary of the results for these related studies is given in Table 1.”

As seen from the text, a table was also added for easier viewing of important data. This allowed for a theoretical comparison (knowledge gap analysis) given in the same section:

“There are a few knowledge gaps that can be identified observing the related work. First, there is a clear lack of application on the specific issue of KXE quality assessment. Second, only a few studies consider the application of existing, tried and tested, CNN architectures and opt for a customized CNN approach. This approach complicates the studies through the introduction of unknown variables, as well as making them harder to reproduce, while they may not provide a benefit. Most of the research does not cross-validate the data, and uses the standard train-test approach. Most authors use images as-is, without attempting to identify if there are performance benefits from using a different size images. It is also important to note the numbers of images used, which range wildly - from less than a hundred images to tens of thousands, with the dataset used in the presented study falling around the middle at 4,000 images. Finally, little testing is done with combinations of various different ML-architectures - despite research in other area showing that these types of methods can yield significantly improved results, only a single paper identified in the related work review applies them for quality assessment [20].”

  • It will be setter to specify the outcomes related to each combination of hyper parametres to specify the best suitable one.

While we agree this would serve to present a more complete view of the study, there are a total of 360 hyperparameter combinations, for three image sizes and five networks, across five different hybrid approach related results. This would require a show of total 27,000 result combinations for a total of six different metrics and standard errors, which is impractical for obvious reasons. Because of this, we believe that it is most practical to present only the best model hyperparameters, as we have.

  • Perform a sensitivity analysis of the model. Specify the model outcome with respect to some other classification criteria such as precision, recall, accuracy, f1-score and Gmean.

The additional metrics were added to the to the results section, in tables three through eigth. To avoid cluttering the section further we have not used the Gmean metric (we assume this was referring to the Geometric Mean), as our datasets are fully balanced.

A brief explanation of the used metrics was added to the Methodology section, as:

“In addition to AUC, a number of different metrics have been used in order to further validate the model. Based on the TP and FN values, along with their accompanying error types - False Positive (FP), and True Negative (TN) values, the following metrics were calculated:

  • Accuracy ACC, calculated as (TP + TN)/(TP + FP + FN + TN)
  • Precision P, calculated as TP/(TP + FP),
  • Recall R, calculated as TP/(TP + FN),
  • Sensitivity Sens, calculated as TN/(FP + TN),
  • Specificity Spec, calculated as TN/(FP + TN), and
  • F1 score, calculated as 2 · TP/(2 · TP + FP + FN).

While the AUC was used as the main metric for model tuning, the additional metrics allow us a more detailed look in the exact nature of the errors our models are producing on the data.”

  • Mention a separate paragraph including the limitation and future work of your study in the conclusion section.

The limitations were addressed in conclusion with the paragraph:

As mentioned, study has certain limitations. For example, there is no way to determine the performance on different XE collection methodologies, as a single source of images was used. Further research on a wider dataset is needed to assess the performance for different XE acquisition methodologies for other diagnostic procedures, such as non-weight bearing XEs and ones without an aluminium calibration disk present.

And the future work was expanded as such:

“As seen for limitations of the study, there is a decent amount of items to be focused on in the future work. Testing and validating the developed methodology on differently collected XE datasets, especially from multiple hospital sources, is key to assure the performance of the developed system in wider use, with the presented work used as a pilot study. Beyond that, development of similar models for different anatomical regions could serve as a useful study. Finally, further application of AI-based methods on collected images in order to automate the system further, such as automatic calculation of adjustment angle could be beneficial, as it would provide more information to XE technicians beyond informing them if the XE performed is satisfactory or not.”

  • In Line 233 , Table ?? will be Table 3.

The wrong reference has been corrected, thank you.

We hope the reviewer will consider the manuscript significantly improved now that their comments have been addressed

Kindest regards,
the authors.

Reviewer 3 Report

The article is related to an interesting issue, i.e. the analysis of x-ray examinations. However, the authors do not undertake any relevant research in it from the perspective of both technology and medicine. Therefore, I have some comments to which please refer.

1. First, the work is not motivated properly. This work seems like a general classification problem for x-rays examinations.

2. What is the knowledge gap closed by this study? The contribution must be clearly stated.

3. The paper lacks a Discussion section. This section is among the most important in articles of an experimental nature. Also, the authors wrote a Conclusion, but it reads like and Summary. This needs improvement. What is sorely lacking there is the author's evaluation of the works read and a word of commentary comparing them with each other. After reading such a discussion, the reader should know which of the presented works is valuable.

4. Improve the conclusions: use the numerical results for experiments to support your claims.

5. Some dissatisfaction is the lack of a valuable literature review. In order to prove the superiority of the proposed approach, I think the author should compare his results with existing methods. And this requires citing those works in the Introduction section.

6. The analysis of the study results is limited to only one evaluation metric (i.e., AUC). And as can be seen from the (very narrow) literature review included in the Introduction section, other authors use metrics such as accuracy, F1 score, and others. The study should be enriched with other evaluation metrics to assess the quality of the studies conducted. In addition, the results should be illustrated with a confusion matrix.

7. At the beginning of the materials and methods section, adding a flow diagram illustrating the entire methodology is worth adding.

8. The authors wrote about the use of 5 neural networks. The authors then described hybrid architectures. How the mentioned neural networks were used in the described architectures is unclear. This explanation should be added.

9. The authors did not present results on test data. What the authors call test data is really validation data because it was used to select the best set of hyperparameters. The dataset should be split into training, validation, and test data. Conclusions should be made based on test data. Making conclusions based on only validation data is the same method error as making conclusions from training data.

Author Response

We would like to thank the reviewer for their detailed and important comments, We sincerely believe they were very important and addressed important issues in our manuscript. We have marked the changes in text made due to this Reviewer’s comments with red text.

  • First, the work is not motivated properly. This work seems like a general classification problem for x-rays examinations. 

We have attempted to make the motivation of the work clearer by including the following paragraph in the Introduction:

“The main motivation of the presented work is to simplify the process of KXE examinations for both the patients and the medical staff. The low quality KXEs refer to those KXEs in which the patient's knee is not positioned properly or is misoriented, not allowing for the medical examiner to diagnose the injury properly. When this happens, a new XE must be performed, which puts undue stress on the hospital system as well as lowering the quality of healthcare obtained by the patient, due to the delay introduced before a diagnosis can be made.”

  • What is the knowledge gap closed by this study? The contribution must be clearly stated.

The knowledge gaps identified by the literature review have been added to the introduction section, under the appropriate subsection “Related work”:

“There are a few knowledge gaps that can be identified observing the related work. First, there is a clear lack of application on the specific issue of KXE quality assessment. Second, only a few studies consider the application of existing, tried and tested, CNN architectures and opt for a customized CNN approach. This approach complicates the studies through the introduction of unknown variables, as well as making them harder to reproduce, while they may not provide a benefit. Most of the research does not cross-validate the data, and uses the standard train-test approach. Most authors use images as-is, without attempting to identify if there are performance benefits from using a different size images. It is also important to note the numbers of images used, which range wildly - from less than a hundred images to tens of thousands, with the dataset used in the presented study falling around the middle at 4,000 images. Finally, little testing is done with combinations of various different ML-architectures - despite research in other area showing that these types of methods can yield significantly improved results, only a single paper identified in the related work review applies them for quality assessment [20].”

The contributions have been clarified by adding the following:

“The contributions of the presented work can be presented as: 

  • Develop an AI-based system for classification of lateral KXE quality, in order to inform the technician performing the examination immediately if the positioning of the KXE is not satisfactory, 
  • Test approaches combining different CNN architectures in various hybrid setups and compare the performance to standard, single-CNN approach in order to determine the need for the use of advanced CNN approaches, and 
  • Determine which parameters of the entire system generate the best performance, in regards to CNN architecture, hybrid combination of the CNNs, hyperparameters, and image size in order to enable simpler future development of models focused on similar issues”
  • The paper lacks a Discussion section. This section is among the most important in articles of an experimental nature. Also, the authors wrote a Conclusion, but it reads like and Summary. This needs improvement. What is sorely lacking there is the author's evaluation of the works read and a word of commentary comparing them with each other. After reading such a discussion, the reader should know which of the presented works is valuable.

The discussion was added as a separate section before the conclusion, as follows.

“The presented results show a certain regularities which will be discussed in this section. Limitations of the study will also be addressed. All of the models have been evaluated using multiple metrics. The main evaluation metric, used in the training procedure, was AUC, with it and Accuracy, F1, Recall, Precision, Sensitivity and Specificity used for model evaluation on five-fold cross validation sets. The result show very low variation in average metric scores between certain metric types, such as AUC, F1 and Accuracy - but the standard variations are somewhat higher with accuracy and F1, compared to AUC -- indicating that AUC is not sufficient as a sole metric to be used for evaluation. This is probably caused by the fact that the models were initially trained using AUC as a main metric, causing a somewhat better average performance across folds. Same can be seen in the case of precision -- specificity, and recall-sensitivity metric pairs, which are equal when rounded to two significant decimals. It must be noted that the part of the reason for this is a relatively small binary test dataset, of only 400 images. Most of the models show higher recall and sensitivity, compared to precision and specificity, which is important to note considering the goal of the system. As this is prevalent across most of the model results, indication is that this is caused by the images contained in the dataset. Image size does not have a large influence on the model performance, with scores for the same network/approach combination being very similar across all image sizes. Here, the indication is that smaller sized images can be used for further development of this and similar systems, due to the simpler handling of such data and faster/less computationally expensive model training on smaller image sizes. First thing to note is the general results of the individual CNN architectures without observing the different approaches. Results show that in most cases more complex networks generate better results, e.g. ResNet152 shows better results than ResNet50, and Xception shows better results than AlexNet. This is not surprising, due to larger models allowing for a higher number of trainable parameters, which means that they can be trained to address more complex problems. Still, this is not always the case, which indicates the possibility of the larger models failing to be trained with the data present in respective folds training/validation set. When the results of different approaches are observed, they show that there is a definite score benefit to the use of a hybrid model which combines the OD CNN with a Classifier CNN. This indicates multiple things. First, due to OD flipping the images improving the results, it can be assumed that the orientation of the XE does play a factor in performance of the classifier. In other words, Classifier networks show better performance in cases where all of the input images are oriented in the same direction. The OD-LFK/RFK approach seems to show a slightly better top performance than OD-LK/RK approach, with top scores around 0.97 when observing AUC for OD-LFK/RFK compared to 0.94 for OD-LK/RK. Within OD-LFK/RFK approach, OD-RFK has slightly superior results, but these fall within the range of standard error on five-fold cross validated dataset, meaning their performance is essentially interchangeable.”

The authors have also moved the subsection “Limitations” to the “Discussion” section as it seemed more appropriate to be placed there, compared to the “Results” section of the manuscript.

  • Improve the conclusions: use the numerical results for experiments to support your claims.

The numerical results for the best models have been added to the conclusions.

“The manuscript presents an automated system for the classification of lateral KXE quality in the acquired dataset. Three separate approaches have been tested, to compare the performance of different combinations of CNNs. When evaluated using different metrics, it can be seen that the best performance is achieved when the ResNet152 is used to discriminate the orientation of XE between left and right, achieving scores of AUC = 0.97 ± 0.09; ACC = 0.97 ± 0.08; F1 = 0.97 ± 0.08; R = 0.99 ± 0.08; P = 0.96 ± 0.06; Sens = 0.99 ± 0.04; Spec = 0.95 ± 0.06, in combination with an Xception CNN as classifier that achieved scores of AUC = 0.97 ± 0.01; ACC = 0.97 ± 0.05; F1 = 0.97 ± 0.05; R = 0.99 ± 0.04; P = 0.95 ± 0.06; Sens = 0.99 ± 0.04; Spec = 0.95 ± 0.06. These results are achieved when combining the networks into an RFK approach - the discriminator detects the orientation of the XE, left oriented images are flipped to match the right image orientation, and then the classifier is applied.”

  • Some dissatisfaction is the lack of a valuable literature review. In order to prove the superiority of the proposed approach, I think the author should compare his results with existing methods. And this requires citing those works in the Introduction section.

A subsection “Related work” was added to the Introduction section, including a brief review of papers with similar goals, as follows:

“Zhang et al. (2021) [13] apply the Feature Extraction CNN on the problem of fetal sonograph quality assessment. On the balanced dataset of 14,700 images, authors manage to achieve AUC and F1 scores of 0.96. Lauermann et al. (2019) [14] described the use of a customized deep learning algorithm, based on CNNs, for the problem of optical coherence tomography angiography quality assessment. The models are evaluated using accuracy and achieve the score of 0.97 on the validation dataset. Jalaboi et al. (2023) [15] focus on the application of a custom CNN architecture "ImageQX". The architecture is trained on 36,509 images, out of which the validation set consists of 9,874 photographs. With the architecture at hand, athors achieve the score of 0.73 ± 0.01. Quality assessment on eye fundus photography is performed by Karlsson et al. (2021) [16 ]. They apply a random forest model on features extracted with Fourier transformation on a publicly available DRIMDB dataset, consisting of 216 images. The authors demonstrate the accuracy score of 0.981, sensitivity score of 0.993 and specificity score of 0.958. Mairhofer et al. (2021) [17]. Authors extract the region of interest and classify the quality into one of three classes using a customized CNN based framework. When evaluating on the set of 950 images, authors achieve the accuracy score of 0.94 with their proposed three-step framework. Chabert et al. (2021) [ 18 ] apply multiple methods (linear discriminant analysis, quadratic linear analysis, support vector machine, logistic regression and multilayer perceptron) for assessment of Lumbar MRI quality. Best results are achieved with a combination of methods, with the recall of 0.82 and AUC of 0.97. Coyner et al. (2019) [ 19] demonstrate the use of a custom deep CNNs for quality assessment of retinopathy images. Using a dataset of 4,000 images authors achieve an AUC of 0.97, with the sensitivity of 0.94 and specificity of 0.84. Czajowska et al. (2022) [ 20 ] apply a hybrid system consisting of a deep CNN and a fuzzy reasoning system on quality assessment of author collected dataset of high-frequency facial ultrasounds. The achieved results show that the proposed system achieves a classification accuracy of 0.92. The summary of the results for these related studies is given in Table 1.”

The comparison to the results was given in the conclusion:

“Comparing the results presented in this study, it is visible that the achieved results are comparable, if not better, then similar approaches performed on different data. Higher sensitivity than specificity shown in presented scores is consistent with previous related studies.”

  • The analysis of the study results is limited to only one evaluation metric (i.e., AUC). And as can be seen from the (very narrow) literature review included in the Introduction section, other authors use metrics such as accuracy, F1 score, and others. The study should be enriched with other evaluation metrics to assess the quality of the studies conducted. In addition, the results should be illustrated with a confusion matrix.

The following metrics were used to further evaluate the models: Accuracy, F1, Precision, Recall, Sensitivity and Specificity. These metrics were added to the Results section in tables three through eigth. Confusion matrix has also been added, but only for the best model for brevity, as it is the most relevant result. The confusion matrix is visible in Figure 8, with the following explanation added:

“To further illustrate the performance of the model as described, a confusion matrix of the entire hybrid approach is shown in Figure 8. A single cross validation fold is randomly selected and used for evaluation, for brevity. It can be seen that on the selected validation set of 800 images (20%), the selected model classifies "acceptable" images as "unacceptable", and 16 "unacceptable" images as "acceptable".”

  • At the beginning of the materials and methods section, adding a flow diagram illustrating the entire methodology is worth adding.

The flow diagram has been added as Figure 1, with the following text added as a description:

“As Figure 1 shows, the methodology consists of three main parts - the dataset collection, model development and model evaluation. The dataset collection was performed by medical experts, and consisted of performing 2,000 KXEs, labeling and splitting them, before converting them to an appropriate format for machine learning methodology to be applied. This was the second step, which mainly consisted of repeating the training process using grid search (GS) hyperparameter variation approach and five-fold cross validation on various image sizes and CNN approaches, which will be described going further. Finally, each of the models was evaluated, with the best performing model being selected for further evaluation based on time performance.”

  • The authors wrote about the use of 5 neural networks. The authors then described hybrid architectures. How the mentioned neural networks were used in the described architectures is unclear. This explanation should be added.

The text has been expanded with the following clarification:

“In other words, this means that each of the CNN architectures as presented in the previous section have been used for each of the separate classifiers presented in the upcoming section. For example, in the approach where the left and right knees are used separately, with the first neural network used to discriminate the orientation and then two networks are used to classify the results as acceptable/unacceptable, each of the aforementioned CNN architectures has been tested for each of the tasks.”

  • The authors did not present results on test data. What the authors call test data is really validation data because it was used to select the best set of hyperparameters. The dataset should be split into training, validation, and test data. Conclusions should be made based on test data. Making conclusions based on only validation data is the same method error as making conclusions from training data.

This was initially poorly explained in the methodology section of the paper. The dataset is indeed split into training, testing, and validation data. For each of the folds, the “train” dataset is split into subsets of training and validation using the default train_test_split available in Scikit-Learn library (75:25), with the separate train data being used to evaluate the model performance as the data that is not seen by the networks during model training for each of the folds individually. In other words, models are fitted using model.fit(train_images, train_labels, validation_data=(validation_images, validation_labels)...). These results weren’t presented as they are not relevant, as you yourself have mentioned, and were only used to monitor for fit performance. We have attempted to make this process more clear by expanding the explanation as such:

“This assures that the dataset is split into training and testing data using an 80:20 ratio split. The training dataset is then split further into training and validation sets using 75:25 ratio (60 % of the total data for training, and 20% for the internal validation). The test data is not used during the fitting process and only evaluated on using trained models, which yields the presented results.”

We hope the reviewer will consider the manuscript significantly improved now that their comments have been addressed.

Kindest regards,
the authors.

Round 2

Reviewer 3 Report

Thank you for addressing my concerns. I accept the paper in its current version.